# Food Security and Foodborne Mycotoxicoses—What Should Be the Adequate Risk Assessment and Regulation?

**DOI:** 10.3390/microorganisms12030580

**Published:** 2024-03-14

**Authors:** Stoycho D. Stoev

**Affiliations:** Department of General and Clinical Pathology, Faculty of Veterinary Medicine, Trakia University, Students Campus, 6000 Stara Zagora, Bulgaria; s_stoev@hotmail.com

**Keywords:** food security, food safety, foodborne ailments, mycotoxins, hygiene control, risk assessment, mycotoxin interaction, control measures, One Health

## Abstract

The purpose of this review is to elucidate the actual threat of the most prevalent mycotoxins in agricultural commodities and human/animal food/feed for the induction of foodborne diseases or ailments. The underestimated hazard of combined mycotoxin uptake by animals or humans is critically discussed with regard to synergistic or additive interaction between some target mycotoxins. The real toxicity of target mycotoxin combinations as it happens in practice is evaluated and possible lower limit values or control measures are suggested in such cases. Some critical points on adequate risk assessment, hygiene control, and regulation of mycotoxins are discussed. The efficiency of current mycotoxin regulations and control measures is evaluated in regard to human/animal health hazards. The risk assessment in the case of multiple mycotoxin exposure of humans/animals via food/feed or agricultural commodities is evaluated and some suggestions are proposed in such cases. Appropriate control measures and food safety issues throughout the food supply chain are proposed in order to prevent the target foodborne diseases. Some preventive measures and possible veterinary hygiene controls or risk evaluations are proposed in some natural cases of foodborne diseases for preventing mycotoxin contamination of animal products designed for human consumption and to avoid possible public health issues.

## 1. Introduction

Mycotoxins are fungal metabolites that are often contaminants of feeds or food commodities. This poses a serious hazard to animal/human health around the world. A lot of health ailments in animals/humans can be provoked by food/feed that is contaminated by mycotoxins. Cereals are invaded by different fungal species in the field or after the harvest, and such invasion and subsequent mycotoxin production is often unavoidable due to environmental predisposing factors such as excessive rain at the time of harvest, increased humidity, and inadequate storage conditions [1,2]. Independent of the large number of natural fungal metabolites (above 400), only 10–14 are mostly responsible for foodborne ailments or compromised public health, e.g., ochratoxin A (OTA), aflatoxins (AFs) among which aflatoxin B1 (AFB1) and aflatoxin M1 (AFM1) are the most problematic, fumonisins (FUMs) among which fumonisin B1 (FB1) is the most problematic, ergot alkaloids, zearalenone (ZEA), patulin (PAT), deoxynivalenol (DON), diacetoxyscirpenol (DAS), T-2, and HT-2. These mycotoxins often contaminate feedstuffs, food commodities, or animal/chick products, e.g., eggs, meat, or milk, in concentrations that can compromise human health or animal wellbeing [3,4,5,6].

Animals exposed to mycotoxins via feedstuffs often have changes in behaviour, such as nervousness or refusal to eat such feedstuffs, showing poor conversion of feed or decreased weight gain. Foodborne diseases, increased number of secondary bacterial infections, or decreased reproductive capacities can be often seen in such animals [7,8]. The well-known toxic effects of mycotoxins are nephrotoxic (mainly OTA and slightly FB1), neurotoxic (mainly FB1), immunosuppressive (mainly OTA, AFB1, DON, and T-2 toxin), genotoxic (mainly AFB1, OTA, T-2), carcinogenic (mainly AFB1, OTA, and FB1), and oestrogenic (ZEA) effects [2,9,10].

Multiple mycotoxin contamination with several mycotoxins often occurs in many feedstuffs or food commodities. The mycotoxin contents in the same cases are often below the maximum permitted levels and within European requirements, but such multiple mycotoxin contamination at low levels could also be harmful to humans or animals, taking into consideration the additive or synergistic interactions between some target mycotoxins, which are often responsible for the appearance of foodborne ailments. Such joint mycotoxin interactions should be carefully investigated via in vitro or in vivo studies and appropriate measures should be taken in regard to the necessary hygiene control and the required risk evaluation. In such a way, the possible hazards for animals or humans could be analyzed in depth, and adequate preventive measures could be proposed for each particular case.

According to the Food and Drug Administration (FDA), the economic costs of crop losses in the USA due to mycotoxin contamination and subsequent condemnation of feeds or food commodities are nearly USD 930 million per year [11]. The Food and Agriculture Organization (FAO) reported that nearly 25% of the world’s crops are contaminated by mycotoxins each year, which contributes to nearly 1 billion tons of annual losses of feeds/foods [12]. However, this percentage greatly underestimates the occurrence above the detectable levels (up to 60–80%), which could be explained by the improved sensitivity of analytical methods and climate change [13]. In this regard, different kinds of losses attributed to mycotoxins are known, e.g., decreased livestock production, illness or death of animals/humans, increased necessity for medical care and veterinary service, increased costs for regulatory and preventive measures or mycotoxins detoxification, economic losses caused by food/feed scrapping, etc. [5,10].

Currently, the European Union (EU) has accepted maximum permissible levels for most dangerous mycotoxins in human food commodities or animal feedstuffs [7], but these limits do not take into consideration joint mycotoxin exposure and mycotoxin interactions, synergistic or additive effects of some mycotoxins, and increased toxicity of such mycotoxin combinations even at lower contamination levels than permitted ones. Therefore, the effectiveness of current regulatory measures could be questionable because the regulations are based only on the toxicity of individual mycotoxins without taking into account possible mycotoxin interactions. In order to ensure adequate food safety and effective control, additional regulatory measures have to be introduced, which must have in mind additive or synergistic mycotoxin interaction. Therefore, the competence of qualified experts in the required research field, including the science of food, veterinary and human medicine, and agriculture, is of crucial necessity for the introduction of such regulatory measures [10].

This review will elucidate the most prevalent mycotoxins in feedstuffs and food commodities and evaluate the hazards of mycotoxin contamination on human or animal health. The possible hazard of joint mycotoxin intake in animals or humans will be briefly elucidated. The effectiveness of the current provisions for the regulation of mycotoxins in feedstuffs and foods will also be discussed in relation to human and animal health. A brief investigation will be made about possible veterinary preventive measures, hygiene control, and risk assessment in regard to some foodborne ailments/diseases provoked by mycotoxins in order to reduce mycotoxin content in meat and other animal products and to prevent the subsequent entering of mycotoxins into commercial channels.

## 2. Mycotoxin Prevalence and Current Regulations

The seasonal weather conditions within each geographical area during the critical growing stages of each plant species are of particular importance for explaining the levels of mycotoxin contamination. The variation in the results is usually a consequence of various important circumstances, such as the type of analyzed samples, the periods of the surveys, and the climatic differences in each particular year of the survey. In addition, various environmental conditions, such as increased humidity, temperature, excessive rainfall, drought conditions, and damage by insects, in addition to the agronomic practices used, could provoke stress and further contribute to the development of mould in the respective plants in the field before harvesting [13,14,15]. For example, the weather conditions known to lead to extensive AFs contamination are mainly increased temperature, scarce rainfall, and drought stress, whereas DOJN and ZEA production by *Fusarium* spp. is mainly facilitated by cool weather and extremely wet growing seasons [13,16]. Some mycotoxins, e.g., OTA and AFs, are mainly produced under the conditions of storage for a prolonged time and, therefore, are known as storage mycotoxins, whereas others, such as FUMs, trichothecenes, DON, ZEA, and some other *Fusarium* mycotoxins are mostly produced in field conditions before harvesting [1]. In addition, the international trade of feed/food ingredients may further facilitate the distribution of materials contaminated with mycotoxins outside their natural areas of occurrence and, therefore, additionally complicate the possible prediction of mycotoxin contamination of feeds or foods.

The range of EU regulatory limits is from 0.1 μg/kg (AFB1 content in processed cereal-based foods for children) up to 4000 μg/kg (FB1 and FB2 content in unprocessed maize for humans). For milk and milk products, the AFM1 limit is 0.05 μg/kg [2,17]. The accepted EU limit of mycotoxins in wheat is 4 μg/kg (AFs), 2 μg/kg (AFB1), 5 μg/kg (OTA), 1250 μg/kg (DON), and 100 μg/kg (ZEA), and the most common mycotoxins in flour prepared from wheat are AFs, OTA, and DON [2,17]. Therefore, the EU limits of the same mycotoxins in processed cereal products are lower, such as 3 µg/kg for OTA, etc. [18]. The maximum permitted levels of mycotoxins in maize when used for human consumption according to EC regulation are 2 μg/kg (AFB1), 10 μg/kg (AFs), 5 μg/kg (OTA), 4000 μg/kg (FUMs), 1750 μg/kg (DON), 350 μg/kg (ZEA), and 100 μg/kg (T-2 + HT-2) [2,7]. The main mycotoxins contaminating rice are AFB1, ZEA, DON, FUMs, OTA, and HT-2/T-2 [19]. The maximum permitted levels of these mycotoxins in rice defined by the EC are 10 μg/kg (AFs), 5 μg/kg (AFB1), 5 μg/kg (OTA), 1250 μg/kg (DON), and 100 μg/kg (ZEA) [2,7].

The most common mycotoxin in barley is DON, whereas AFs and DON prevail in cereal porridge, and AFs prevail in breakfast cereals [5]. The most common mycotoxins in vegetables and fruits are PAT, OTA, and trichothecenes [5,20]. AFs were most often found as natural contaminants in South Asia (78% positive samples with an average contamination level of 128 μg/kg), followed by Southeast Asia (55% positive samples with an average contamination level of 61 μg/kg) [21]. In Germany, seven oilseed samples investigated in 2010 were found to contain AFB1 above the maximum limit [20]. ZEA was the most often contaminant in North Asia (56% positive samples with an average level of 386 μg/kg). DON was the most often found natural contaminant in North Asia (78% positive samples with an average contamination level of 1060 μg/kg). However, the highest average DON contamination was found in North America (68% positive samples with an average contamination level of 1418 μg/kg) [21] (Table 1). DON was also equally prevalent in food from Europe and Canada with about 57% of the European [22] and about 59% of the Canadian food [23]. In Austrian feeds and feed raw materials, DON was found in around 60% of investigated cereal samples other than maize and in around 95% of maize samples [20]. High DON contamination levels were also found in liquid pig feed samples in The Netherlands, and 10% of the same sample exceeded the maximum permitted values [24]. DON is also the most prevalent mycotoxin in beer and, consequently, would be a real public health problem [25]. AFs, FUMs, OTA, and ZEA have also been reported to contaminate the beer at various stages of brewing [25]. FUMs were found to be the most frequent contaminant in South America (77% positive samples with an average contamination level of 2691 μg/kg). OTA was seen to be most prevalent contaminant in South Asia (55% positive samples with a mean level of 20 μg/kg). OTA was also seen to be a frequent contaminant in Eastern European samples (49% positive) evaluating the OTA exposure of the EU population, but the average level of contamination was much lower (4 μg/kg) [6,21] (Table 1 and Table 2). OTA in the same studies was found to be most prevalent in cocoa products (81%), dried fruit (73%), and wine (59%), but it was most prevalent in red or sweet wine as compared to the other wines [6,26]. In addition to cocoa products, coffee and chocolate were also reported to contain high contamination levels of the same mycotoxin [27,28,29]. A high prevalence of OTA was also found in wheat-based products (94%) in Canada [23], but the highest OTA levels have been reported in countries in Southern Europe [26]. AFs were found to be the most common mycotoxins in the peanuts and pistachios [5].

OTA, in addition to AFs, has also been reported to contaminate animal products such as dried meat and other meat products such as sausages and salami or eggs, which presents a global problem for human health [32,33,34]. DON and ZEA were also reported to contaminate meat, but to a lower degree [5]. AFs have been reported to be the most important contaminants of milk or dairy products, e.g., yogurt and cheese [5]. The most often contaminants of eggs have been reported to be AFB1, OTA, ZEA, and DON [35], which appear to be a potential public health problem.

Contamination peaks of mycotoxins are often traceable to target regions and are usually seen in response to extreme weather conditions. On the other hand, having in mind climate change in Europe and all over the world, a possible increase in the magnitude or frequency of human/animal exposure to mycotoxins is expected to occur, which could further increase public health concerns. In this regard, mycotoxins that are not usually found in foods/feeds from European countries might occur as a result of changes in the distribution of some target fungal species in regions with climate changes, e.g., wider dissemination of *Fusarium* fungi and mycotoxins is expected to be seen in EU countries [36,37,38,39]. Therefore, a different exposure pattern to mycotoxins is expected to occur in EU countries now or in the near future. For example, a strong *Aspergillus flavus* infection was seen in 2003 because of the hot and dry weather, which led to high contamination of maize with AFB1 in northern Italy [40]. A study of 110 samples revealed the presence of AFB1 in 75% of them, with a mean level of 4.4 μg/kg. Because of the use of such maize as a feed source for dairy cattle, high contamination of AFM1 was subsequently seen in milk. Therefore, thousands of tons of milk had to be removed due to exceeding the EU maximum permitted levels of 0.05 μg/kg [20,41]. In addition, some species, such as *F. verticillioides*, usually encountered in warmer and drier regions of Europe, e.g., Spain and Italy, were the most prevalent *Fusarium* species in German maize in 2006, which subsequently increased FUMs contamination of maize to 34% of the studied samples [42]. Such high contamination levels of AFB1 in the EU also shows that climate change will entail a change in the pattern of the current mycotoxin distribution in the future.

In this regard, the development of predictive models for mycotoxin contamination in cereals, foods, and feedstuffs based on the data of the regional climate would be useful to evaluate the risk of mycotoxin contamination in each season. Although the climate is one of the most influential parameters in regard to the extent of mycotoxin contamination, some other measures such as crop rotation, tillage, or planting time (earlier planting of maize is important) are also of crucial importance in order to reduce mycotoxin contamination of cereals [20].

In order to obtain reliable results for mycotoxin contamination, proper sampling has to be performed. It is well known that sampling could be a significant source of error in quantifying mycotoxin contamination levels due to difficulties in sampling from large batches of grain and because of the different levels of mycotoxin contamination in various places in a single feed/food ingredient [43]. In order to ensure an effective sampling procedure for cereal mycotoxin detection or quantification, EC Regulation 2023/2782 defines the sampling and analyzing methods for mycotoxin control in food/feed and repeals EC Regulation No 401/2006. In this regard, the details of sampling methods, acceptance parameters, and defined analytical criteria for the methods used are provided for the official controls, in addition to the criteria for reporting and interpretation of the results received [44]. In this regard, a lot of analytical methods for defining mycotoxin content in food commodities or feed ingredients were elaborated, e.g., immunoassay, high-performance liquid chromatography (HPLC), gas chromatography (GC), tandem mass spectrometry (MS/MS), gas chromatography/mass spectrometry (GC/MS) and liquid chromatography/mass spectrometry (LC/MS), among which the last one is increasingly widespread for detection of multiple mycotoxin conjugates [45,46,47,48,49,50,51,52]. Some high or ultra-performance chromatography systems, together with mass spectrometry, are also found to be useful in the examination of multiple mycotoxins in various foods/feeds [53]. The areas that need further investigation and refinement are regarding the conjugated or modified mycotoxin determination and the elaboration of a new, convenient, rapid, and cheap analytical approach [54]. Moreover, further studies on the elaboration and application of such methods are necessary. Some analytical methods, such as commercial ELISA kits, are also very useful because they are cheap and easy to apply [54].

## 3. Joint Mycotoxin Exposure as a Cause of Foodborne Ailments

Currently, there is evidence of mycotoxin involvement in diseases in humans or farm animals, such as pulmonary oedema in pigs, equine leukoencephalomalacia, vulvovaginitis or rectal prolapse in pigs, Alimentary Toxic Aleukia in people, stachybotryotoxicosis, mycotoxic porcine/chicken nephropathy, Balkan Endemic Nephropathy (BEN), ergotism, and some others [7,10,55,56,57]. Different animals have different sensitivity to mycotoxins, as poultry species are less sensitive to the toxicity of FUMs, DON, and ZEA, but pigs are more sensitive to T-2 and DON [54].

Unfortunately, the toxic effects of mycotoxins, and especially the toxicity of various mycotoxin combinations on human health, are scarcely investigated. Currently, there is no sound evidence for the involvement of mycotoxins in some particular diseases in humans from developing countries where people are continuously exposed to mycotoxin-contaminated foods. However, a simple connection was seen between the FB1 content in maize, the quantity of ingested maize products by humans, and the rate of oesophagal cancer in people, which suggested that FB1 is probably responsible for human oesophagal cancer in some countries, e.g., South Africa and China [58]. It was also found that pregnant women who ingested high concentrations of FUMs via their food at the initial stage of the pregnancy have a high risk of the appearance of neural tube defects, such as birth defects of the spinal cord or brain in their newborn children [59]. Idiopathic Congestive Cardiopathy (ICC) in humans is another common disease seen in South Africa, which is also associated with ingestion of high levels of FB1 and other trichothecenes, incl. moniliformin (MON), which is suspected to be partly responsible for the cardiac weakness. The disease is mainly established in elderly people who consumed a lot of home-produced maize and drank a lot of homemade beer [7,60].

On the other hand, feedstuffs contaminated by mycotoxins contribute to mycotoxin contamination of some food products from animal origin, e.g., milk, dairy products, meat, or eggs, due to the transmission of some mycotoxins from the forages to defined food commodities. This situation may further contribute to the increase in human exposure to mycotoxins [4,61,62]. Another circumstance contributing to mycotoxin exposure in humans is their thermal and chemical stability and the minimal loss during thermal treatments or production processing [63].

The multiple mycotoxin contamination of forages and food commodities was reported to provoke many foodborne ailments/diseases in animals and humans (Figure 1) [7,56,57,64]. In addition, mycotoxins are reported to be responsible for some secondary bacterial diseases due to their immunosuppressive effects [56,57]. Some foodborne mycotoxicoses, e.g., equine leukoencephalomalacia, vulvovaginitis and rectal prolaps in pigs, Alimentary Toxic Aleukia in people, porcine pulmonary oedema, human oesophagal carcinoma, stachybotryotoxicosis (Figure 1), mycotoxic porcine/chicken nephropathy (Figure 2), ergotism, and many other diseases or ailments in animals or humans are some of the well-known examples of foodborne mycotoxicoses [2,10].

*Fusarium* mycotoxins, such as DON and ZEA, are mainly responsible for multiple mycotoxin contamination of maize and, less often, of oats, barley, and wheat. ZEA is reported to contaminate cereal products, including feedstuffs, pasta, bread, and beer [65], as well as animal products, including milk, meat, and eggs [5]. DON is mainly reported to contaminate wheat, rye, corn, oats, barley, rice, and sorghum [5]. ZEA was found to be involved in the appearance of many vulvovaginitis, rectal or vaginal prolapse in female pigs (Figure 1), and some other estrogenic symptoms, including swelling of the mammary glands or infertility [7,66], whereas in male pigs, a feminization and decrease in testosterone levels and/or spermatogenesis, as well as decreased libido, were observed [67]. ZEA has a genotoxic action and could be partly responsible for breast and esophageal carcinomas [68,69]. The International Agency for Research on Cancer (IARC) classified ZEA in Group 3, mycotoxins that do not exert carcinogenic effects on humans [70,71]. Hyperestrogenism in young female pigs is defined by vaginal or rectal prolapses, and a high percentage of vulvovaginitis in the same pigs, known to be the main symptoms of ZEA-toxicosis, is usually seen only after prolonged ZEA ingestion above a month. However, the first clinical symptoms of ingestion of such feedstuffs moulded by *F. culmorum* or *F. graminearum* are due to the toxic action of DON, e.g., cytotoxic effect on neurons (manifested by paresis), damages in the gastrointestinal system (manifested by vomiting), and immunosuppression (manifested by secondary microbial infections) [2,7,72,73]. In humans, the main clinical symptoms due to DON exposure are vomiting, acute nausea, diarrhea, abdominal pain, dizziness, headache, and fever [5].

Some other *Fusarium* mycotoxins, such as DAS, T-2, and HT-2, which are produced mainly by *Fusarium* spp. belonging to the *Sporotrichiella* section or by *F. poae* species, were reported to also possess strong cytotoxic, genotoxic, and immunosuppressive effects [74]. Barley and oats are mainly the cultures that are frequently contaminated with T-2 and HT-2 [20]. Damage to the hematopoietic system, damage to eggshells and egg production, feed refusal, and growth retardation are the main symptoms of T-2 toxicity [75]. Damage to the cardiovascular system, growth retardation, and lung damage are the main symptoms of DAS toxicity, which could add to the clinical picture provoked by T-2 and HT-2 [7]. The most sensitive species to this fusariotoxicosis are pigs and poultry [20], and the symptoms are seen mainly after ingestion of hay, straw, or grain wintered outdoors. This fusariotoxicosis in people is known as Alimentary Toxic Aleukia [76]. The clinical signs characteristic of this fusariotoxicosis are catarrhal or haemorrhagic gastroenteritis accompanied by ulcerations and necrotic changes in the gastrointestinal system, damage in the kidney, heart, liver, peripheral ganglia, and brain, responsible for subsequent muscular spasms, paresis of limbs, and tremors [7]. Abdominal pain, diarrhea, nausea, tremors, and weight loss are seen in the initial stages of this fusariotoxicosis [77].

Equine leukoencephalomalacia is another important foodborne mycotoxicosis, which is provoked mainly by FUMs, among which FB1 is the most toxic. These mycotoxins were reported to contaminate mainly maize in developing countries [55,78]. Porcine pulmonary oedema, which was recognized for the first time in the USA and is responsible for the death of many pigs, is another foodborne mycotoxicosis provoked by mouldy maize containing FUMs [79]. The heart failure and subsequent oedema in the lungs of pigs can be explained by disturbances in the contractility of the myocardium in pigs, induced by the increase in sphingosine and subsequent inhibition of L-type calcium channels in the myocardium [80]. A similar disturbance in the contractility of the myocardium in humans is known as ICC and was first recognized in 1980 in S. African rural hospitals. The same disease was supposed to be due to the intake of *Fusarium* mycotoxins such as FB1, MON, and some others [1,7].

AFs are potent mycotoxins that are often seen in feedstuffs or food commodities together with other mycotoxins, which can strongly complicate the clinical and pathological findings. These mycotoxins, among which AFB1 is the most dangerous, affect mostly the liver; the most sensitive species are turkeys and ducks, as well as all young animals. The typical signs in the early stage of intoxication are fatty changes and necrotic damage in the liver, accompanied by connective tissue proliferation, enlargement of the gall bladder, and intestinal damage. The typical signs in later stages are icterus and cirrhosis of the liver, hydrothorax, and ascites, accompanied by skin thickening near the mouth or neck and papillomatous formations on the mucosal surface of the abomasus, which are characteristic symptoms mainly in cattle. A decrease in body weight gain, immunosuppression, and anaemia [7,62,81,82] are some additional chronic signs of aflatoxicosis in animals, poultry, or humans, which can be complicated by some other co-contaminating mycotoxins with similar toxicity [7]. AFB1, together with other mycotoxins, can also induce oedematous changes and is often associated with Kwashiorkor disease in humans [71,83]. AFB1 is classified by the IARC as a carcinogenic mycotoxin for humans (Group 1 mycotoxin) and is the main cause of nearly 28% of all carcinomas in the liver [71]. AFs were also reported to have teratogenic effects on embryos and to be able to cross the placental barrier [84].

Mycotoxic porcine nephropathy (MPN) is also a mycotoxicosis, which is induced by combined mycotoxin action, as reported in some Balkan or African countries, with OTA, FB1, and penicillic acid (PA) being the most important mycotoxins involved [56,57]. The same mycotoxins have synergistic (OTA and PA) [85,86] or additive (OTA and FB1) interactions [87]. The main clinical symptoms of MPN are strong damage to the kidneys (Figure 2) [56,57,88,89], but a decrease in weight gain, nervous symptoms (Figure 3), hepatocellular changes, and egg weight reduction (Figure 4) were also seen in laying hens or chicks exposed to the same mycotoxins [90,91,92,93,94].

The fungal species *Claviceps purpurea*, which can produce a lot of mycotoxins, e.g., ergocristine, ergocryptine, peptide alkaloids such as ergosine, lysergin derivatives such as ergine and ergometrine, ergotamine and ergosecaline, biogene amines such as histamine and acetylcholine and some others, can provoke another dangerous mycotoxicosis known as ergotism [7]. These mycotoxins are found in the sclerotia of this fungus, which is similar to a dark, big wheat grain [3]. This fungus contaminates mainly wheat, rye, barley, oats and millet. In humans, ergotism is famous as St Anthony’s fire, which was accompanied by hallucinations and was the cause of death for many humans in France in the past [95]. Nowadays, St Anthony’s fire is still found in some developing countries [96,97]. The main pathological/clinical symptoms of ergotism are ischemic necroses of peripheral parts of the body, e.g., tail, ears, or crown of hooves, induced by the contraction of vessels, gangrene of the peripheral part of extremities, enhanced contractions of the uterus and subsequent prolapse of uterus and/or abortions, as well as some gastrointestinal signs [96,97,98].

**Figure 2 microorganisms-12-00580-f002:**
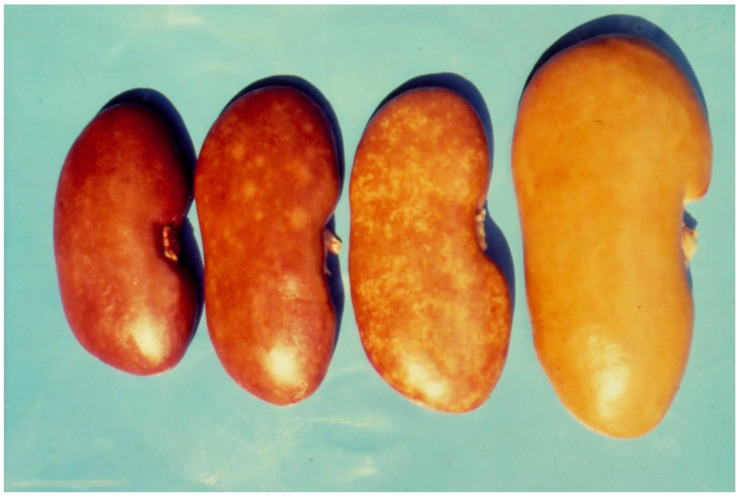
Macroscopic changes in kidneys with mycotoxic nephropathy. Different extents of enlargement and mottled or pale surface of kidneys in pigs aged between 6 and 8 months originated from slaughterhouses [56,88].

**Figure 3 microorganisms-12-00580-f003:**
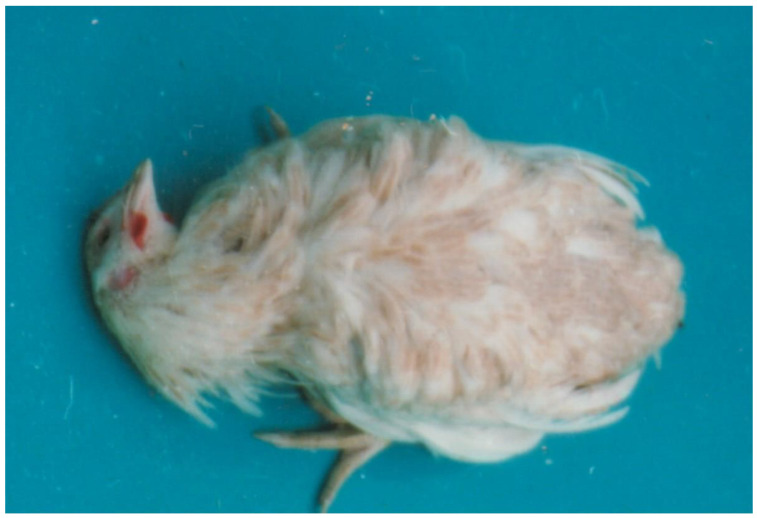
Nervous signs, e.g., torticollis in a bird exposed to a mouldy diet contaminated with 790 µg/kg OTA and 2000–5000 µg/kg PA during a period of 70 days [86].

**Figure 4 microorganisms-12-00580-f004:**
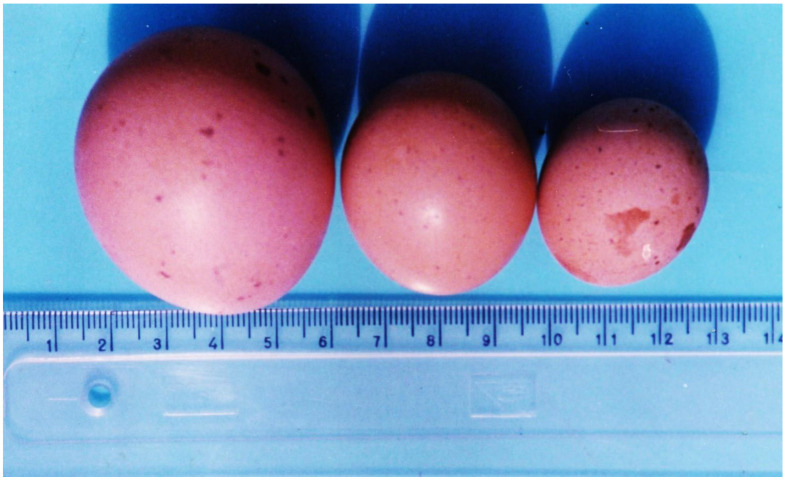
Small size of eggs with a weight of 15.8 g (**centre**) and 25.8 g (**right**) and different-sized damages or spots in the shell originated from laying hens treated with 5 mg/kg OTA in the diet. Normal size of an egg from the control group (**left**) [94].

Another mycotoxicosis widely known in animals is stachybotryotoxicosis, which is induced by the fungal species *Stachybotrys altra* (*Stachybotrys alternans*), encountered in moist straw/oats/hay and other cellulose-rich foods. This fungus can produce very toxic compounds, e.g., satratoxins, roridins, and verrucarins, and can be defined by its black colour. These fungal mycotoxins irritate the mucosa of the oral cavity and gastrointestinal tract, causing strong hyperaemia and inflammation. On the other hand, after mycotoxins’ accumulation, deep neurotrophic symmetrical necrotic damages and ulcers on the mucosa of the gastrointestinal system, oedema, and haemorrhages are seen [7,99,100] (Figure 1). This mycotoxicosis shows a stationarity because the fungus survives in the soil (outside the contaminated materials) for a lot of time, which explains the repeated contamination of animal feedstuffs [7].

Another dangerous mycotoxin for humans is PAT, which is often seen together with some other mycotoxins. PAT is reported in apples, grapes, and pears damaged by brown rot, and also in the juice from the same fruits. It is also reported to contaminate vegetables, cereals, and various types of cheese [5]. The rotten part of fruits should be removed before consumption to reduce the PAT content ingested by consumers. PAT was reported to induce various toxic effects, e.g., neurotoxic, cytotoxic, or carcinogenic, in addition to reproductive disturbances [83,101,102]. The cytotoxic effect is manifested by damage to the gastrointestinal tract, liver, and kidneys, and disturbances in the endocrine and immune systems [103]. PAT is classified by the IARC as a suspected carcinogen (from Group 3) [104,105].

It is worrying that a lot of mycotoxins have carcinogenic (Figure 5), genotoxic, teratogenic (Figure 6), and immunosuppressive properties in addition to their acute toxic action [68,106,107,108,109,110,111,112]. A good example in this regard is FB1, which is suspected to provoke human esophageal cancer in South Africa [58] and to induce liver carcinomas in rats [113], in addition to its nephrotoxic effect [56,57]. The available data in the literature investigated mostly the carcinogenic effect of single and, rarely, double mycotoxin exposure via in vitro studies [114], but in vivo studies investigating the chronic effect of multiple mycotoxins on the induction of neoplasia are scarce [115,116,117].

It is well known that most mycotoxins have strong immunosuppressive properties and can increase susceptibility to secondary bacterial infection in concentrations similar to those in practice [118,119,120,121,122,123], including susceptibility to salmonellosis [124,125,126] or colibacillosis [127] or provoke a heavy progression of some infections such as *Pasteurella multocida*-induced disease [128] or porcine reproductive and respiratory syndrome (PRRS) [129,130,131,132] (Figure 7) or parasitic invasions such as coccidiosis [92,133]. Having in mind this circumstance, it can be assumed that combined mycotoxin exposure could compromise the immune system of animals in very low concentrations, having in mind the synergistic or additive interaction between some target mycotoxins [87,90]. Therefore, it could be concluded that the increased morbidity and mortality in livestock and poultry exposed to various mycotoxin combinations via feedstuffs is possibly a consequence of increased susceptibility to secondary microbial infections or a heavier course of some parasitic or bacterial diseases [2]. Oxidative stress, which can be provoked by many mycotoxins, could additionally threaten animal and human health [134,135].

## 4. Some Critical Points in Adequate Risk Assessment, Hygiene Control, and Regulation of Mycotoxins

The following components of the analysis of risk, e.g., assessment, communication, and management of risk, should be analyzed to provide adequate food safety. In order to provide adequate risk assessment, the following components have to be evaluated: identification and characterization of hazards, characterization of possible risks, and assessment of the exposure. To ensure adequate risk communication, a regular exchange of knowledge and opinions between the risk evaluators, risk managers, and consumers is necessary during analyzation of the risk. However, the adequate risk management decision should be based not only on the risk assessment but some additional matters should also be addressed, e.g., economic, environmental, ethical, and other circumstances, in addition to the feasibility of effective control [1,7].

Therefore, the Joint FAO/WHO Expert Committee on Food Additives (JECFA), after evaluating the most dangerous mycotoxins, provided a mechanism for assessment of the toxic impact of each mycotoxin [136], which includes defining the “no observed effect level” (NOEL) via experimental investigations and subsequently applying “a factor of safety”, in order to estimate the Provisional Tolerable Daily (PTDI) or Weekly Intake (PTWI). This approach evaluates the maximum tolerated mycotoxin levels in feedstuffs or food commodities but could not be used in regard to the carcinogenic effect of some mycotoxins such as AFS, where the “as low as reasonably achievable” (ALARA) approach or “as low as possible but technologically feasible and analytically detectable in the food ready for consumption” approach should be applied [137]. The IARC provides such a mechanism for the assessment of cancerogenic properties of mycotoxins, according to which mycotoxins were divided into Group 1 (“cancerogenic mycotoxins for humans”, incl. AFB1), Group 2A (“probably cancerogenic mycotoxins for humans”), Group 2B (“possibly cancerogenic mycotoxins for humans”, incl. FB1 and OTA), and Group 3 (“not classifiable as cancerogenic for humans”) [70]. Some revisions of the accepted criteria for this classification and regular updates in regard to some mycotoxins are periodically undertaken [10,71]. The European Food Safety Authority (EFSA) also ensures independent scientific advice regarding food-related risks in relation to AFs via using a margin of exposure (MOE) approach. More than 200,000 analytical studies on Afs presence were used in the assessment. When using such a MOE approach for the characterization of risk for the incidence of hepatocellular carcinomas (HCC) in male rats after AFB1 exposure, “a benchmark dose lower confidence limit” (BMDL) for a benchmark response of 10% of 0.4 µg/kg b.w. per day was established. Unfortunately, the establishment of such BMDL via using the available data for humans was found to be inappropriate. A potency factor of 0.1 in relation to AFB1 was applied in such assessment in regard to AFM1 [138].

In regard to DAS, T-2, and HT-2, a group TDI of 25 ng/kg b.w. for a single mycotoxin or mycotoxins combination was recently decided by JECFA [139], and the former group “provisional maximum tolerable daily intake” (PMTDI) of 60 ng/kg b.w. (incl. T-2 and HT-2), accepted on the 56th meeting and updated on the 83rd meeting (via the inclusion of DAS), was changed. A PMTDI of 0.4 µg/kg b.w. was also set in regard to PAT (Table 3) [17,41,139,140,141,142].

Similarly, a group TDI value of 1 µg/kg b.w. was designed in regard to DON and its derivatives, e.g., 15-acetyldeoxynivalenol (15-ADON), 3-acetyldeoxynivalenol (3-ADON), and plant metabolite DON-3G, which was based on experiments in mice with chronic mycotoxin exposure [142] (Table 3). In acute cases, however, 8 µg/kg b.w. per eating occasion was accepted as a reference dose due to gastrointestinal damage reported in DON-exposed humans in China [142,143]. This TDI was based on the circumstance that the same three derivatives can be biotransformed into DON in humans [144]. The evaluation of the risk of DON exposure to humans in the EU population revealed that a part of it is exposed to concentrations that suppose a significant health hazard [65,145].

The TDI of FUMs was set at 2 µg/kg b.w. because FB1 was suspected to provoke neural tube defects in the embryo [146] and scarce data are available for the mechanism of renal excretion of FB1 in humans. The same mycotoxin is scarcely absorbed in the gastrointestinal system, is quickly eliminated from the circulated blood by hepatobiliary excretion, and is mainly eliminated by the faeces [65], which explains the absence of the required attention from the scientific community. Additional efforts to clarify the toxicokinetics profile of this mycotoxin after oral ingestion would be useful to contribute further to better risk assessment [65].

The TDI of AFs (a total sum of all forms of aflatoxins) is the lowest one, similar to the maximum permitted levels of AFs in feeds or food commodities (Table 3) because AFB1 is classified by the IARC as a Group 1 human carcinogen [70,71]. AFs are eliminated via the feces, milk, and urine, but the same mycotoxins can also be found in animals’ organs, meat, or chicken eggs and, therefore, represent a real health hazard. Moreover, the contamination of cereal-based products, in addition to peanut cake, palm kernel, corn gluten meal, pork products, milk, and eggs, increases the possibility of human exposure to AFs [20,65]. It is a worrying circumstance that in non-ruminant animals, more than 80% of AFs are absorbed via the gastrointestinal tract, mostly by using passive transportation, in comparison to the low rate of absorption of some other dangerous mycotoxins such as OTA or FUMs (from 1% up to 60%) [147].

The TDI of OTA, similar to the TDI of Afs, has the lowest value (0.0002–0.017 µg/kg b.w.), as the lower dose concerns its carcinogenic effect (Table 3). OTA contamination mainly occurs in cereals, e.g., barley, wheat, maize, and rye, in addition to some poultry or animal products such as meat-based products, kidneys, and eggs [7]. The enterohepatic circulation of OTA contributes to its retention for a longer time in the gastrointestinal system, which could aggravate the health hazard. The proposed JECFA initial value of 112 ng/kg b.w. PTWI for OTA corresponded to about 16 ng/kg b.w. PTDI [148]. This PTWI was then decreased to 100 ng/kg b.w., corresponding to 14 ng/kg b.w. PTDI [149], but after that, it again increased to 120 ng/kg b.w. or nearly 17 ng/kg b.w. PTDI (Table 3). The PTWI assessment is based on the nephrotoxicity of OTA without consideration of its carcinogenicity. There is another calculation of TDI made by Kuiper-Goodman and Scott, which takes into consideration the cancerogenic effect of OTA, and such calculation ranged between 0,2 and 4,2 ng/kg b.w, depending on the methodology used [150]. Having in mind both calculations of TDI, the calculated average daily intakes of OTA for people living in BEN-endemic regions in Bulgaria (26.8 ng/kg b.w. in 1988, 36.4 ng/kg b.w. in 1989, and 34.2 ng/kg b.w. in 1990, respectively) [3,7] are greatly above the TDI of 17 ng/kg b. w. (in regard to the nephrotoxic effect of OTA), and even more strongly above the TDIs taking into consideration the cancerogenic property of OTA [150].

In regard to possible veterinary hygiene control in OTA content in animal products, the introduced measures in some EU countries, such as Denmark, would not be able to provide adequate food safety because they are not quite appropriate [3]. The accepted regulation in Denmark requires the study of all “mottled and/or enlarged kidneys” for OTA content during the slaughtering of pigs and condemnation of the carcasses if OTA content is more than 10 µg/kg [151]. Such regulations, however, are not very relevant and satisfactory because the mottled appearance of kidneys can be provoked only after prolonged OTA exposure of nearly 1–3 months [85,152]. Therefore, such regulation cannot provide OTA-free pork and cannot restrict OTA-contaminated pork from moving through commercial channels, which poses a potential risk to human health [3,153]. A possible and easily achieved control measure would be to study a few blood samples from pigs or poultry in farms with nephropathy problems a few weeks (for pigs) or a few days (for poultry) before the slaughter time. In these cases, the feed supply could be changed with a more relevant one for a week (in pigs) or for several days (in poultry), if OTA is present in the blood. A possible approach to prevent OTA contamination of meat and derived products could be extending the fasting period (feed deprivation) just before slaughter time [153,154]. Such a measure is easy to perform and very effective due to the short half-life of this mycotoxin in pigs (72–120 h) and especially in poultry (4 h) [155]. In such cases, the OTA levels in the blood or tissues of the respective poultry and pigs will be strongly decreased, and any losses due to the scrapping of pig/chicken meat will be avoided. Such measures could ensure a more effective control for restricting subsequent OTA intake by humans via pork products as compared to the toxicological studies of “mottled kidneys” according to the regulations in Denmark. If the same control measures cannot be performed, the removal (condemnation) of the kidneys and liver in already-slaughtered poultry and only the kidneys in already-slaughtered pigs, where the largest quantity of OTA accumulates, would be enough [154].

It is of crucial importance that the HACCP (Hazard Analysis and Critical Control Point) system is introduced worldwide to ensure the regular identification and assessment of possible hazards at different stages of food/feed production. The subsequent undertaking of adequate measures for ensuring regular control and food/feed safety is also required, e.g., prevention strategies, good manufacturing practices, and regular control at various stages of food/feed processing or production from the harvest of raw ingredients up to the end user [3]. The knowledge of the content of target mycotoxins such as DON, ZEA, and FUMs in each step of processing maize and cereals is of crucial importance due to the high contamination levels of these mycotoxins in raw ingredients. Therefore, the enforcement of regular surveillance control and adequate food safety regulations is also of critical necessity to provide safe food/feed and to decrease incidences of foodborne ailments. In this regard, automated sorting and segregation are applied for the separation of AFs-contaminated peanuts, and cleaning cereals prior to milling is applied to remove spores of fungi, debris, and broken grains containing high concentrations of mycotoxins [156,157]. The removal of bran from flour intended for bread might also decrease mycotoxin intake by humans, and weaker constraints on raw materials should be adopted. Nevertheless, it is debatable whether the consumer would prefer the bread and pastry prepared from wholemeal with its known health benefits or white bread and pastry without bran in order to decrease the risk of mycotoxin content [18].

The monitoring of food/feed quality and application of mycotoxin regulations is mainly available in developed countries, whereas such standards are not available or are ineffective in developing countries. The introduction of such regulations in developing countries is often very complicated because of problems with the food supply. Moreover, these regulations often promote the export of the best quality crops to comply with the regulations, which could increase the risk of mycotoxin exposure and health ailments in local people due to the circumstance that foods/feeds or food ingredients with low quality usually remain for local consumption.

Currently, a lot of countries have elaborated their own regulations designed to ensure effective control of mycotoxin contamination in feeds or food ingredients [2,7]. Nowadays, introducing worldwide legislation and internationally recognized regulations is of crucial importance for minimizing the exposure of humans/animals to various mycotoxins when the risk assessment of such exposure is significant. However, in the process of evaluation of each particular risk assessment, the toxic effect of each mycotoxin and mycotoxin combination should be taken into consideration, in addition to the estimated mycotoxin exposure of animals or humans. Simultaneous intake of several mycotoxins via feedstuffs or food commodities, although at very low concentrations for a long period (such as simultaneous ingestion of OTA and PA), is of crucial importance for the appearance of some foodborne ailments and could be a significant risk for animal/human health. Therefore, the real toxic and carcinogenic effects of various target mycotoxin combinations, which are often seen in real practice, should be carefully evaluated, and new limit values must be introduced in such cases. Unfortunately, the current Maximum Permitted Levels (MPLs) and TDIs values of mycotoxins (Table 3) as accepted in the EU for animal feed [140,158,159,160] or human food [17,140] are not very reliable because the same take into consideration only the known toxic effect of each particular mycotoxin or a group of similar mycotoxins (such as the sum of AFs or T-2 + HT-2 + DAS or FB1 + FB2) on different animal species, but do not address the actual mycotoxin interactions (synergistic or additive) as it happens in real practice. The United States Department of Agriculture has also accepted similar limits in the USA, but they also neglected the actual mycotoxin interactions as they occur in practice [161]. Therefore, additional regulations and control measures should be introduced in such cases, which are based on the known synergistic or additive interactions of some mycotoxins and their stronger toxic effects on animals/humans in such cases, e.g., OTA and PA [85,86,162] or OTA and FB1 [87], in order to provide adequate risk assessment and food safety. The necessity of international harmonization of such regulations and control measures should also be undertaken to facilitate global food trade and food safety. The same regulations and risk assessments should be carefully designed and based on the toxic effects of multiple mycotoxin exposure as occurs in real practice [10]. Such MPLs and TDIs should also be based on extensive studies in order to prevent excessive restrictions and excessive economic loss [1,7].

The elaboration of regulatory measures for mycotoxin content in food and feed that are internationally recognized is a very difficult task, and, therefore, preliminary elaboration and introduction of some temporary indicative limits in cases that pose a significant danger to human health would be a more useful and easy task to achieve.

The development of a suitable networking system for the dissemination of important knowledge should also be introduced and sustained at the international level, e.g., staff training at regional and international levels. Such international regulations and control measures should be scientifically based and elaborated using agreement between all stakeholders, e.g., manufacturers, consumers, policymakers, and traders, to ensure widespread distribution and compliance with the same rules. In the process of elaboration of such international regulations and standards, a lot of circumstances should be taken into consideration, e.g., scientific validity, adequate risk assessment, analytical accuracy, and establishing the toxicity of mycotoxin combinations, which most often occurs in the field, in addition to target economic factors, including the commercial interests of the countries and the need for food delivery in order to avoid unjustified rejections of raw food ingredients and possible economic difficulties for producers [1,7].

## 5. Concluding Remarks

It is well known by the scientific community that human/animal exposure to mycotoxins via food/feed cannot be fully prevented since mycotoxins are natural contaminants of food/feed ingredients. Some mycotoxins, e.g., aflatoxins, zearalenone, and ochratoxin A, are much more hazardous due to their passage into the milk of lactating cows (parent toxins or their metabolites, e.g., aflatoxin M1, zearalenone, α zearalenol), eggs, and meat (e.g., ochratoxin A).

The current national rules and regulations for monitoring and control of mycotoxins in food commodities and feedstuffs are mostly based on the evaluation of the threat of each individual mycotoxin for each individual country. Now, it is crucial to introduce carefully designed surveillance control and modern internationally recognized biomonitoring measures to evaluate animal/human exposure to mycotoxins. Such control measures should be implemented globally for reliable control of factors that compromise the quality of feed and food ingredients and the commodity system. It is important to highlight that introducing too many restrictive food safety regulations could lead to unjustified rejections of some raw food ingredients and the respective commodities, which could have fatal consequences for some small producers or traders and put unjustified barriers in international trade [1]. Therefore, synchronizing existing national regulations and developing international regulations and standards for acceptable content of mycotoxins or target mycotoxin combinations in foods/feedstuffs and raw ingredients should be undertaken. Such international regulations would significantly improve the protection of all consumers worldwide and facilitate trade at the international level, as well as food safety based on the latest scientific advances and adequate risk assessment.

The novelty of this review paper is the evaluation of risk assessment of combined mycotoxin exposure to some target mycotoxin combinations and the proposal for the elaboration of new regulatory measures in such cases, which have to take into account additive or synergistic interaction between the same mycotoxins as happen in the real practice. Nowadays, the regulations and standards in the EU and US do not take into consideration mycotoxin interaction and the combined toxicity or carcinogenicity of mycotoxins, and they are based only on their individual toxic effects. Therefore, the combined toxicity of some target combinations of mycotoxins must be deeply examined due to synergistic or additive interactions between mycotoxins and should be taken into consideration for regulatory purposes. Some additional experimental studies in animals or humans designed to clarify relationships between target mycotoxins and the respective health outcomes, e.g., neural tube defects, idiopathic congestive cardiopathy, or oesophagal cancers in humans, are also crucial. The mycotoxin exposure of humans/animals in some countries, e.g., in the Balkan countries, is often seen to be below the TDI for each separate mycotoxin, but the joint toxic action can often greatly exceed the toxic effects of all the individual mycotoxins [2].

The elaboration of a suitable networking system for the worldwide dissemination of target knowledge should also be introduced and sustained at the global level. Collaboration between research teams, consumers, producers, traders, and policymakers is also crucial to solving food safety issues and to facilitate the wide distribution and compliance with the same rules and standards in order to avoid unjustified rejections of raw food ingredients and possible economic difficulties for producers. Some economic and political factors, e.g., how to ensure sufficient food supplies to the countries concerned, as well as some commercial issues, should also be taken into consideration in the decision-making process. Any effort to improve the quality of food commodities must also be reconciled with people agreeing to bear any associated increase in the price of the feed or food concerned.

## Figures and Tables

**Figure 1 microorganisms-12-00580-f001:**
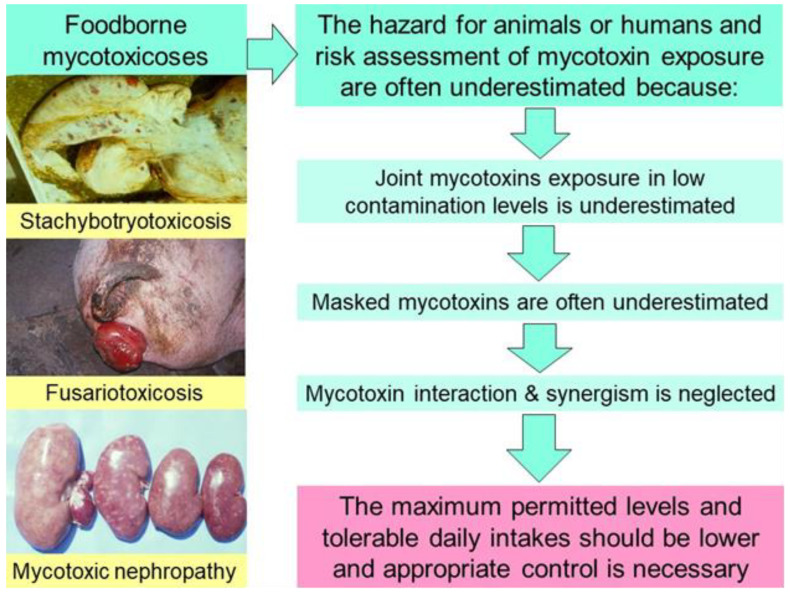
Foodborne diseases due to underestimated hazards from combined mycotoxin exposure, e.g., synergistic mycotoxin interaction, masked mycotoxins, and neglected hygiene control [7].

**Figure 5 microorganisms-12-00580-f005:**
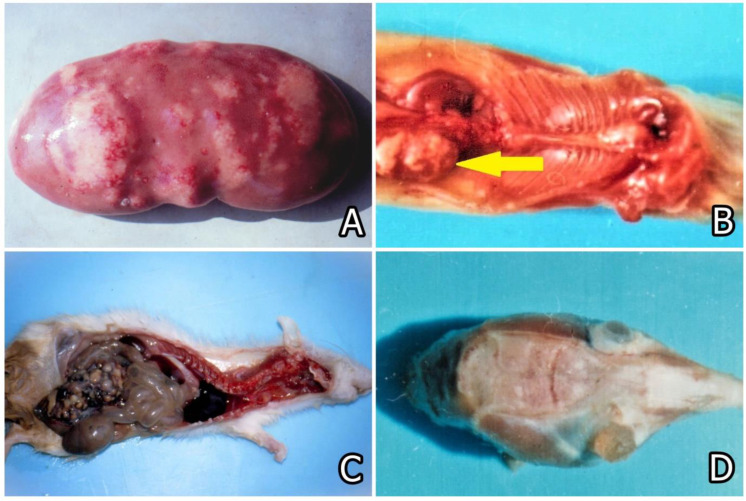
(**A**) Neoplasia (fibroadenoma and fibroma) in spontaneous cases of MPN in Bulgaria [88]; (**B**) A kidney with adenocarcinoma (pale protruding neoplastic areas on kidney surface) in a rat treated with 5 mg/kg OTA via the consumed feed during an experimental period of 24 months [108,109]; (**C**) Intestine with adenocarcinoma (large protruding pale neoplasia on intestinal surface) in a rat treated with 10 mg/kg OTA via the consumed feed during an experimental period of 19 months [108]; (**D**) An eye with squamous cell carcinoma in a rat treated with 10 mg/kg OTA via the consumed feed during an experimental period of 24 months [108].

**Figure 6 microorganisms-12-00580-f006:**
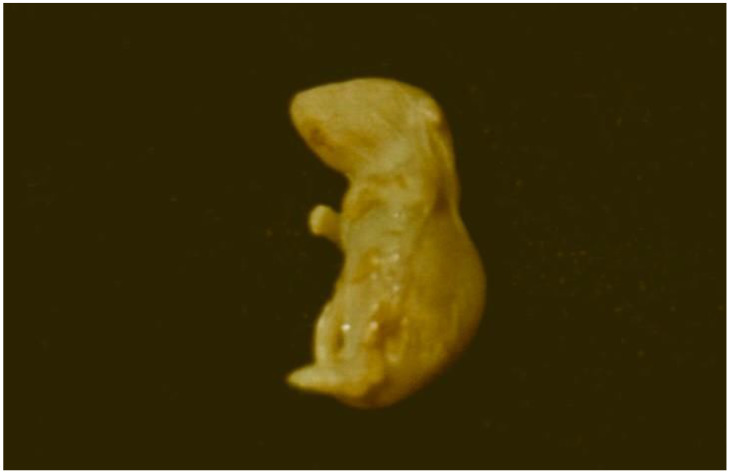
Some deformities (malformations), e.g., anophthalmia, astomia, and peromelia in the right extremities in a newborn mouse born from a mother treated with 20 mg/kg OTA in the consumed feed between days 7 and 12 of gestation [110].

**Figure 7 microorganisms-12-00580-f007:**
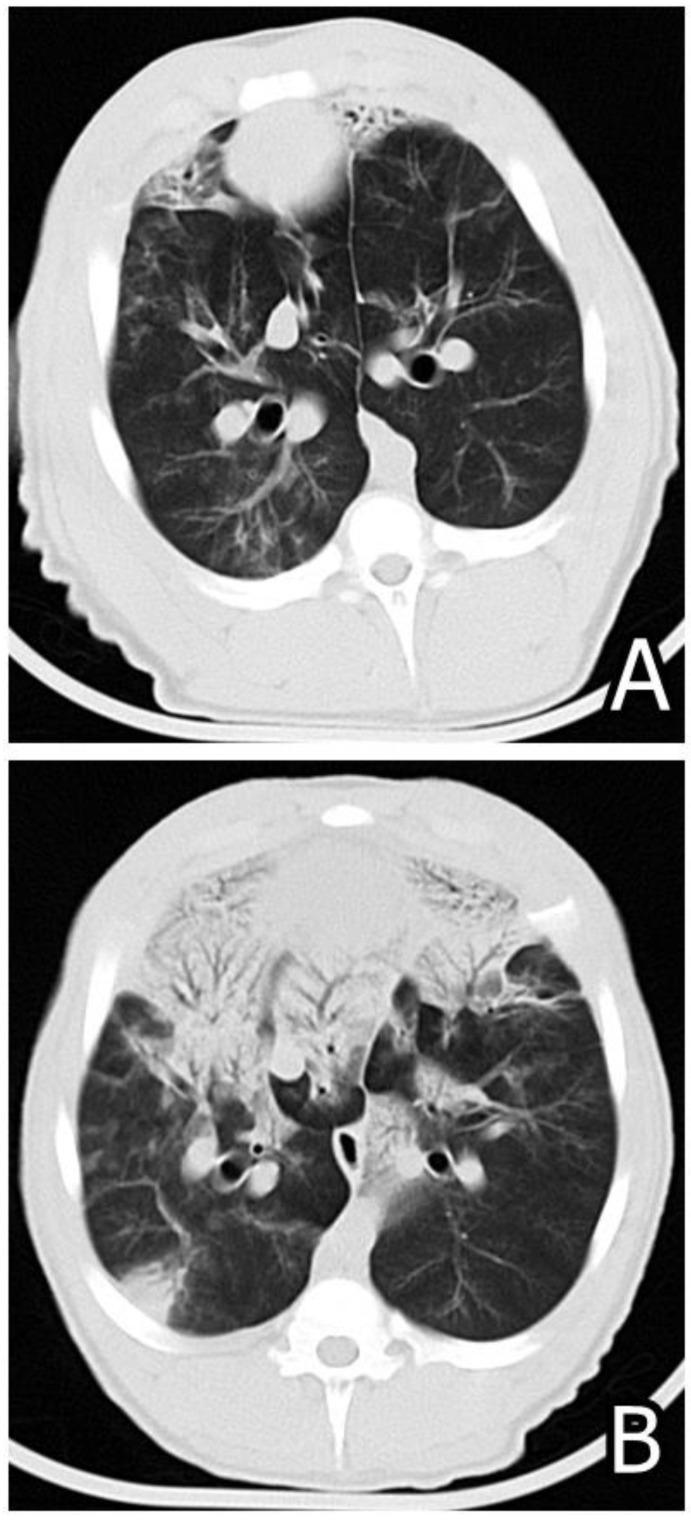
(**A**) Computed tomography (CT) photo of lung damages in *M. hyopneumoniae* experimentally compromised pig in a definite sectioning plane done on day 58 of the experiment showing small focal damages with patchy ground glass opacification. (**B**) CT photo of the lung damages in *M. hyopneumoniae* experimentally compromised pig treated additionally with 20 mg/kg FB1 via the consumed feed conducted in the same sectioning plane on day 58 of the experiment showing severe worsening of pneumonic changes as seen from the enlargement of the same damage [131,132].

**Table 1 microorganisms-12-00580-t001:** A brief overview of mean natural mycotoxin contamination of feed worldwide.

Mycotoxin	Feeds or Ingredients	Mean Level (μg/kg)	Positive %	Continent (Region)	Reference
AFs	various	128	78	South Asia	[21]
AFs	various	61	55	Southeast Asia	
OTA	various	20	55	South Asia	
OTA	various	4	49	Eastern Europe	
FUMs	various	2691	77	South America	
DON	various	1418	68	North America	
ZEA	various	386	56	North Asia	
DON	various	1060	78	North Asia	

**Table 2 microorganisms-12-00580-t002:** A brief overview of ranges of natural mycotoxin contamination of feed or feed ingredients in Europe.

Mycotoxin	Feeds or Ingredients	Range (μg/kg)	Number of Samples	Positive %	Continent (Region)	Reference
OTA	wheat, maize	22–33	82	2	Europe	[30]
FB1	wheat, maize	36–5114	82	44	Europe	
ZEA	wheat, maize	58–387	82	15	Europe	
DON	wheat, maize	74–9528	82	63	Europe	
HT-2	wheat, maize	22–116	82	9	Europe	
AFs	various	0.5–66	127	25	Southern Europe	[31]
OTA	various	1–54	46	22	Southern Europe	
FUMs	various	25–36,390	89	66	Southern Europe	
DON	various	52–4827	348	66	Southern Europe	
ZEA	various	10–2939	303	28	Southern Europe	
T-2/HT-2	various	35–137	65	8	Southern Europe	

**Table 3 microorganisms-12-00580-t003:** Tolerable Daily Intake (TDI) of mycotoxins according to EU regulations and recommendations [17,41,139,140,141,142].

Mycotoxins	TDI (µg/kg b.w.)
Ochratoxin A (OTA)	0.0002–0.017 (the lower figure refers to a cancerogenic effect)
Aflatoxins (AFs) (sum of AFB1, AFB2, AFG1, AFG2, AFM1)	No exact value (ALARA principle is applied)—less than 0.001–0.01 are advisable—the lower figure refers to a cancerogenic effect
Patulin (PAT)	0.4
Deoxynivalenol (DON)	1
Nivalenol (NIV)	1.2
T-2 + HT-2 toxins + DAS	0.025
Zearalenone (ZEA)	0.25
Fumonisins (FUMs)	2

## Data Availability

Not applicable.

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
