# Peer review of "Food Security and Foodborne Mycotoxicoses—What Should Be the Adequate Risk Assessment and Regulation?"

_microorganisms, 2024, doi:10.3390/microorganisms12030580_

Round 1

Reviewer 1 Report

Comments and Suggestions for Authors

This review aimed to elucidate the actual threat of the most prevalent mycotoxins in agricultural commodities and human/animal food/feed for appearance of foodborne ailments and diseases.

General comments

-        The work is well organized and contributes significantly to the field

-        This manuscript requires an English editing.  

Minor comments

Line 22: please add One Health in keywords

Line 35-36: please rewrite the sentences and avoid to repeat the most dangerous.

Line 62: is nearly should be are nearly

Please revise all abbreviations; add the full name in the first mention, then abbreviate all over the manuscript.. eg. EU (line 69)..

Line 127, 129, 133, 138, 144, 275, 300: delete found to be

Please write the name of fungus such as Fusarium, A. flavus, F. verticillioides in italic

Line 222: in is repeated and correct diseases

Line 245: delete could

Line 271: replace found to be with are mainly

Line 274, 277, 411: write including as full word not abbreviate

Line 422: replace (make worse the health of animals or humans) with (threat animal and human health)

Line 540-545: the paragraph is very long please divide into short sentences to be easily read.

This review aimed to elucidate the actual threat of the most prevalent mycotoxins in agricultural commodities and human/animal food/feed for appearance of foodborne ailments and diseases.

General comments

-        The work is well organized and contributes significantly to the field

-        This manuscript requires an English editing.  

Minor comments

Line 22: please add One Health in keywords

Line 35-36: please rewrite the sentences and avoid to repeat the most dangerous.

Line 62: is nearly should be are nearly

Please revise all abbreviations; add the full name in the first mention, then abbreviate all over the manuscript.. eg. EU (line 69)..

Line 127, 129, 133, 138, 144, 275, 300: delete found to be

Please write the name of fungus such as Fusarium, A. flavus, F. verticillioides in italic

Line 222: in is repeated and correct diseases

Line 245: delete could

Line 271: replace found to be with are mainly

Line 274, 277, 411: write including as full word not abbreviate

Line 422: replace (make worse the health of animals or humans) with (threat animal and human health)

Line 540-545: the paragraph is very long please divide into short sentences to be easily read.

Comments on the Quality of English Language

-        This manuscript requires an English editing.  

Author Response

A DETAILED ANSWER TO REVIEWER 1

The option “track changes” is now used to highlight all corrections in the manuscript.

-This manuscript requires an English editing.

-Answer – English editing is now prepared. Some grammatical and spelling/typographical errors are also corrected.

Line 22: please add One Health in keywords

-Answer – “One Health” is now added in keywords

Line 35-36: please rewrite the sentences and avoid to repeat the most dangerous.

-Answer – The repetition of most dangerous is now avoided as referee suggested

Line 62: is nearly should be are nearly

-Answer – This mistake is now corrected accordingly “is nearly” to “are nearly”

Please revise all abbreviations; add the full name in the first mention, then abbreviate all over the manuscript.. eg. EU (line 69).

-Answer – All abbreviations are now given with their full name in the first mention and then abbreviated all over the manuscript

Line 127, 129, 133, 138, 144, 275, 300: delete found to be

-Answer – “found to be” is used because these investigations are not mine and it is not quite correct to use simple past tense in such cases. Now, the same sentences are corrected in such a way, which allow omission of “found to be”.

Please write the name of fungus such as Fusarium, A. flavus, F. verticillioides in italic

-Answer – All names of fungi are now given in italic.

Line 222: in is repeated and correct diseases

-Answer – It is now corrected as appropriate.

Line 245: delete could

-Answer – “could” is now deleted

Line 271: replace found to be with are mainly

-Answer – “found to be” is now removed and the same sentence is corrected as appropriate.

Line 274, 277, 411: write including as full word not abbreviate

-Answer – “incl” is now given as full word “including” without abbreviation.

Line 422: replace (make worse the health of animals or humans) with (threat animal and human health)

-Answer – It is done as suggested

Line 540-545: the paragraph is very long please divide into short sentences to be easily read.

-Answer – It is done accordingly and the same sentence is now separated to two different sentences.

Reviewer 2 Report

Comments and Suggestions for Authors

Dear authors,

I have reviewed in detail the manuscript entitled “Food Security and Foodborne Mycotoxicoses – What Should be the Adequate Risk Assessment and Regulation?” The purpose of this review is to elucidate the actual threat of the most prevalent mycotoxins in agricultural commodities and human/animal food/feed for appearance of foodborne ailments and diseases.

I believe it has been a good review and aligns with the objectives of the publication

Line 13: Write correctly in English: “mycotoxins”

Line 44: Write correctly  in English: “ infections”

Line 62: There are more recent articles indicating that this percentage of crop contamination (25%) is higher.

Line 103: Please write “Fusarium” in italics

Line 174: Please write “Fusarium” in italics

Line 177: Please write “A. flavus” in italics

Line 178: Maybe there is a mistake un the word: “weater”

Line 183: Please write “Fusarium verticillioides” in italics

Line 184: Please write “Fusarium” in italics

Line 201: There is new regulation. Regulation (EU) 2023/2782 of 14 December 2023 laying down the methods of sampling and analysis for the control of the levels of mycotoxins in food and repealing Regulation (EC) No 401/2006.

Line 222: There is some kind of typographical error in the word: “diseases”

Line 241: There is some kind of typographical error in the word: “trichothecenes”

Line 318:  Please write “Fusarium” in italics and capital letter.

Line 346: Please use the units instead of ppb

Line 349: Please use the units instead of ppm

Line 395, Line 409 and Line 423: Figure 5, 6 and7. Please write the units in these paragraphs instead of ppm.

Line 451: I thinkyou should discuss MOEs in relation to aflatoxins

Line 466: Maybe in PAT there is a provisional Tolerable intake, please, check it (PMTDI: 0.4 ug/kg bw)

Author Response

A DETAILED ANSWER TO REVIEWER 2

The option “track changes” is now used to highlight all corrections in the manuscript. Some grammatical and spelling/typographical errors are also corrected.

Line 13: Write correctly in English: “mycotoxins”

-Answer: The word mycotoxins is now corrected

Line 44: Write correctly  in English: “ infections”

-Answer: The word “infectctions” is now corrected to “infections”

Line 62: There are more recent articles indicating that this percentage of crop contamination (25%) is higher.

-Answer: This suggestion of the referee is now taking into account. The following discussion and a new reference is now added: “However, this percentage greatly underestimates the occurrence above the detectable levels (up to 60–80%), which could be explained by a combination of the improved sensitivity of analytical methods and impact of climate change [13].”

-New reference:”Eskola, M., Kos, G., Elliott, C., Hajšlová, J., Mayar, S., Krska R. (2020) Worldwide contamination of food-crops with mycotoxins: Validity of the widely cited ‘FAO estimate’ of 25%. Crit. Rev. Food Sci. Nutr, 60, 16, 2773-2789, DOI: 10.1080/10408398.2019.1658570” 

Line 103: Please write “Fusarium” in italics

-Answer: “Fusarium” is now given in italics.

Line 174: Please write “Fusarium” in italics

-Answer: “Fusarium” is now given in italics.

Line 177: Please write “A. flavus” in italics

-Answer: “A. flavus” is now given in italics.

Line 178: Maybe there is a mistake un the word: “weater”

-Answer: This spelling error is now corrected “weater” to “weather”

Line 183: Please write “Fusarium verticillioides” in italics

-Answer: “Fusarium verticillioides” is now given in italics.

Line 184: Please write “Fusarium” in italics

-Answer: “Fusarium” is now given in italics.

Line 201: There is new regulation. Regulation (EU) 2023/2782 of 14 December 2023 laying down the methods of sampling and analysis for the control of the levels of mycotoxins in food and repealing Regulation (EC) No 401/2006.

-Answer: Thanks a lot for this update. It seems another regulation is accepted during the preparation of this review paper. Now it is included as appropriate in the sentence: “Therefore, in order to ensure effective sampling procedure for cereal mycotoxin detection or quantification the EC Regulation 2023/2782 is laying down the methods of sampling and analysis for the control of the levels of mycotoxins in food and repealing EC Regulation No 401/2006.”

The same is replaced in the Reference as well:

“[42] EC Regulation No 2023/2782 of 14 December 2023 laying down the methods of sampling and analysis for the control of the levels of mycotoxins in food and repealing EC Regulation No 401/2006, Off. J. Eur. Union L., 1-44, http://data.europa.eu/eli/reg_impl/2023/2782/oj

Line 222: There is some kind of typographical error in the word: “diseases”

-Answer: This typographical error is now corrected “diseses” to “diseases”

Line 241: There is some kind of typographical error in the word: “trichothecenes”

-Answer: This typographical error is now corrected “trichotecens” to “trichothecenes”

Line 318:  Please write “Fusarium” in italics and capital letter.

-Answer: “Fusarium” is now given in italics and with capital letter.

Line 346: Please use the units instead of ppb

-Answer: It is done “ppb” is now given as “µg/kg”

Line 349: Please use the units instead of ppm

-Answer: It is done “ppm” is now given as “mg/kg”

Line 395, Line 409 and Line 423: Figure 5, 6 and7. Please write the units in these paragraphs instead of ppm.

-Answer: It is done accordingly as referee suggested

Line 451: I think you should discuss MOEs in relation to aflatoxins

-Answer: MOEs are now discussed in relation to AFs. The following discussion and a new reference is now added: “The European Food Safety Authority (EFSA) also provides independent scientific advice on food-related risks in relation to AFs via using a margin of exposure (MOE) approach. More than 200,000 analytical results on the occurrence of AFs were used in the evaluation. A benchmark dose lower confidence limit (BMDL) for a benchmark response of 10% of 0.4 µg/kg b.w. per day for the incidence of hepatocellular carcinomas (HCC) in male rats following AFB1 exposure was set by using such a MOE approach. However, the calculation of a BMDL from the human data was found as inappropriate. For AFM1, a potency factor of 0.1 relative to AFB1 was used in this assessment. MOE values for AFB1 exposure ranged from 5,000 to 29 and for AFM1 from 100,000 to 508. The calculated MOEs are below 10,000 for AFB1 and also for AFM1, which raises a health concern [136].

-New reference: “[136] EFSA CONTAM Panel (EFSA Panel on Contaminants in the Food Chain) (2020). Scientific opinion – Risk assessment of aflatoxins in food. EFSA Journal, 18(3), 6040, pp 1-112, https://doi.org/10.2903/j.efsa.2020.6040”

Line 466: Maybe in PAT there is a provisional Tolerable intake, please, check it (PMTDI: 0.4 ug/kg bw)

-Answer: It is now added in the sentence “A PMTDI of 0.4 mg/kg b.w. was also set for PAT”

Reviewer 3 Report

Comments and Suggestions for Authors

The review article “Food Security and Foodborne Mycotoxicoses – What Should be the Adequate Risk Assessment and Regulation?” is a very interesting review about the problem of the presence of mycotoxins in agricultural commodities and in human and animal foods or feed.

In my opinion, the paper submitted is well written, easy to read and its structure is adequate for a review work.

I have some suggestion to improve this manuscript:

Major comments:

-I suggest explaining the criteria used for references selection as keywords, years, reference types.

-Summarize in a table lines 127-188. I suggest adding a table containing data and references about the mycotoxins found (type of mycotoxin, % and type of feed/food contaminated, Region/ country where the contaminated foodstuffs were found …). In my opinion a table might help the reader better understand the distribution of mycotoxins.

-I suggest emphasizing, in abstract or/and in conclusion, the novelty of this scientific review comparing to the other similar existing reviews.

Minor comments:

-Please, write Fusarium in italics and with the first capital letter (lines 103, 107, 174, 184, 318, 715, 752)

-Please, write the complete name of Aspergillus flavus and write it in italics (line 177). Aspergillus appears in the text for the first and only time in this sentence.

-Please, write F. verticillioides in italics (line 183)

-Please, write correctly trichothecenes (lines 108, 241)

-Please, delete “or poultry” (line 226)

-Please delete the repetition “in” (line 222)

-Please, correct “couuld” (line 245)

-I suggest rewrite this sentence “This circumstance could further contribute to the increase of mycotoxins exposure of humans” for example “This situation may further contribute to the increase in human exposure to mycotoxins.” (lines 248, 249)

-Please, write “thermal treatments” instead of “termal”

-Please, write Fusarium graminearum and Fusarium culmorum in abbreviated form “F. graminearum, F. culmorum).

-Please, correct “inflmation”

-References should be written according to the Microorganism journal “Instructions for Authors” . Please, see https://www.mdpi.com/journal/microorganisms/instructions

Journal Articles: 1. Author 1, A.B.; Author 2, C.D. Title of the article. Abbreviated Journal Name Year, Volume, page range.

Author Response

A DETAILED ANSWER TO REVIEWER 3

The option “track changes” is now used to highlight all corrections in the manuscript. Some grammatical and spelling/typographical errors are also corrected.

I suggest explaining the criteria used for references selection as keywords, years, reference types.

-Answer: There are not any permanent criteria used for references selection such as keywords, years, reference types, etc. Sometimes, such references were sent by some research organizations such as “Academia.edu”, which know my field of interests, but sometimes such information was sent by various journals, used for publication of my papers such as Toxins, Crit. Rev Food Sci Nutr, etc. The content of some other papers is well known due to my previous investigations. Only part of the papers were found due to my target investigations in the literature using some keywords such as food security; food safety; foodborne ailments; mycotoxins; hygiene control; risk assessment; mycotoxin interaction; control measures, etc. The same keywords are given below the abstract.

Summarize in a table lines 127-188. I suggest adding a table containing data and references about the mycotoxins found (type of mycotoxin, % and type of feed/food contaminated, Region/ country where the contaminated foodstuffs were found …). In my opinion a table might help the reader better understand the distribution of mycotoxins.

-Answer: This suggestion was really difficult to fulfill, because many heterogeneous pieces of information had to be integrated in one table such as mean level or range of mycotoxin contamination, different feed ingredients or feedstuffs or grain or food products or animal products, etc. Moreover, if such information is added for all EU countries and for various kind of feed ingredients or grain or various food commodities, such a table will be very large and a lot of additional references have to be added. Such a work requires some furtherl investigations and a lot of time and efforts. Regardless these enormous difficulties, 2 different tables are now elaborated to fulfill this requirement of the Referee, but the same tables are simplified as much as possible to make this elaboration possible. These tables describe “A brief overlook of mean natural mycotoxin contamination of feed worldwide (Table 1)” and “A brief overlook of ranges of natural mycotoxin contamination of feed or feed ingredients in Europe (Table 2)”

I suggest emphasizing, in abstract or/and in conclusion, the novelty of this scientific review comparing to the other similar existing reviews.

-Answer: The novelty of this scientific review is now emphasized in a more appropriate way in the Concluding remarks of the paper according to referee suggestions as follow: „The novelty of this review paper is the evaluation of risk assessment of combined mycotoxin exposure to some target mycotoxin combinations and the proposal for elaboration of new regulatory measures in such cases, which have to take into account additive or synergistic interaction between the same mycotoxins as happen in the real practice. Nowadays, the regulations and standards in EU and US do not take into consideration mycotoxin interaction and the combined toxicity or carcinogenicity of mycotoxins, and the same are based only on individual toxic effects of mycotoxins. Therefore, the joint toxicity of some target combinations of mycotoxins should be deeply examined due to synergistic or additive interactions between mycotoxins and should be taken into consideration for regulatory purposes. Some additional experimental studies in animals or humans designed to clarify relationships between target mycotoxins and the respective health outcomes (e.g., Neural Tube Defects, Idiopathic Congestive Cardiopathy or esophageal cancers in hu-mans) are also of crucial importance. The human- or animal mycotoxin exposure in some countries (e.g. in the Balkan countries) is often seen to be within the TDI for each separate mycotoxin, but the joint toxic effect may often exceed strongly the toxicity of all individual mycotoxins [2].”  

Please, write Fusarium in italics and with the first capital letter (lines 103, 107, 174, 184, 318, 715, 752)

-Answer: It is now done accordingly as referee suggested.

Please, write the complete name of Aspergillus flavus and write it in italics (line 177). Aspergillus appears in the text for the first and only time in this sentence.

-Answer: It is now done accordingly as referee suggested.

Please, write F. verticillioides in italics (line 183)

-Answer: It is now done accordingly as referee suggested.

Please, write correctly trichothecenes (lines 108, 241)

-Answer: This typographical error is now corrected accordingly as referee suggested.

Please, delete “or poultry” (line 226)

-Answer: It is now done accordingly as referee suggested and “or poultry” is deleted.

Please delete the repetition “in” (line 222)

-Answer: The repetition of “in” is now deleted.

Please, correct “couuld” (line 245)

-Answer: This typographical error is now corrected accordingly as referee suggested and the same word is even removed as unnecessary.

I suggest rewrite this sentence “This circumstance could further contribute to the increase of mycotoxins exposure of humans” for example “This situation may further contribute to the increase in human exposure to mycotoxins.” (lines 248, 249)

-Answer: It is now done accordingly as referee suggested and the sentence “This circumstance could further contribute to the increase of mycotoxins exposure of humans” is now replaced with the sentence “This situation may further contribute to the increase in human exposure to mycotoxins.”

Please, write “thermal treatments” instead of “termal”

-Answer: It is now done accordingly as referee suggested and the “thermal treatments” is used instead of “termal” 

Please, write Fusarium graminearum and Fusarium culmorum in abbreviated form “F. graminearum, F. culmorum).

-Answer: It is now done accordingly as referee suggested and Fusarium graminearum and Fusarium culmorum are given in abbreviated form “F. graminearum, F. culmorum”.

Please, correct “inflmation”

-Answer: This spelling error is now corrected accordingly - “inflammation” instead “inflmation”.

References should be written according to the Microorganism journal “Instructions for Authors” . Please, see https://www.mdpi.com/journal/microorganisms/instructions

Journal Articles: 1. Author 1, A.B.; Author 2, C.D. Title of the article. Abbreviated Journal Name Year, Volume, page range.

-Answer: The References are now written according to the Microorganism journal “Instructions for Authors” as referee suggested.